

# Modelling the Impact of Palaeogeographical Changes on Weathering and CO2 during the Cretaceous-Eocene Period

Nick R. Hayes[1,2], Daniel J. Lunt[3], Yves Goddéris[4], Richard D. Pancost[1,5], and Heather Buss[1]

[1]School of Earth Sciences, University of Bristol, Wills Memorial Building, Queens Road, Bristol, UK, BS8 1RJ
[2]Environment Agency, Horizon House, Deanery Road, Bristol, UK, BS1 5TL
[3]School of Geographical Sciences, University of Bristol, University Road, Bristol, UK, BS8 1SS
[4]Géosciences Environnement Toulouse, CNRS—Université Paul Sabatier - IRD, 31400 Toulouse, France
[5]Organic Geochemistry Unit, School of Earth Sciences, Cabot Institute for the Environment, University of Bristol, BS8 1RJ, UK

**Correspondence:** Nick R. Hayes (Nicholas.Hayes@environment-agency.gov.uk)

**Abstract.** The feedback between atmospheric $CO_2$ concentrations and silicate weathering is one of the key controls on the long term climate of the Earth. The potential silicate weathering flux (as a function of conditions such as temperature, runoff, and lithology), or "weatherability", is strongly affected by continental configuration, and thus the position of continental landmasses can have substantial impacts on $CO_2$ drawdown rates. Here, we investigate the potential impact of palaeogeograpical changes

on steady-state $CO_2$ concentrations during the Cretaceous-Eocene period (145-34 Ma) using a coupled global climate and biogeochemical model, GEOCLIM, with higher resolution climate inputs from the HadCM3L General Circulation Model (GCM).

We find that palaeogeograpical changes strongly impact $CO_2$ concentrations by determining the area of landmasses in humid zones and affecting the transport of moisture, that runoff is a strong control on weatherability, and that changes in weatherability

could explain long term trends in $CO_2$ concentrations.As Pangaea broke up, evaporation from the ocean increased and improved moisture transport to the continental interiors, increasing runoff rates and weathering fluxes, resulting in lower steady-state $CO_2$ concentrations. Into the Cenozoic however, global weatherability appears to "switch" regimes. In the Cenozoic, weatherability appears to be determined by increases in tropical land area, allowing for greater weathering in the tropics.

Our modelled $CO_2$ concentrations show some strong similarities with estimates derived from proxy sources. Crucially,

we find that even relatively localised changes in weatherability can have global impacts, highlighting the importance of so-called weathering "hot-spots" for global climate.Our work also highlights the importance of a relatively high-resolution and complexity forcing GCM in order to capture these hot-spots.



## 1 Introduction

One of the key controls on the long-term climate of the Earth is the concentration of atmospheric $CO_2$. Throughout Earth's history, concentrations of $CO_2$ in the atmosphere have changed substantially, for example, from several thousand ppm in the early Phanerozoic to under 200 ppm during the last glacial maximum (e.g. Royer, 2006; Bouttes et al., 2011; Foster et al., 2017). Atmospheric $CO_2$ concentrations are ultimately determined by the balance of $CO_2$ emitted from volcanic sources (volcanic degassing), and the drawdown of $CO_2$ by the chemical weathering of silicate rocks (Berner et al., 1983). This "simple" balance
is complicated by the complexity of chemical weathering processes, especially in the geological past.

Chemical weathering rates of silicate minerals are sensitive to climate variables such as temperature and precipitation in addition to intrinsic factors (i.e. factors inherent to the rocks themselves, such as lithology) (Strakhov, 1967; Walker et al., 1981; Berner et al., 1983; White and Blum, 1995; Oliva et al., 2003; West et al., 2005). While there is evidence from individual sites that weathering is accelerated under warm, humid conditions (e.g. White and Blum, 1995), it is unclear whether such a
trend can be applied at the global scale. Evidence from field and modelling studies of weathering profiles suggests a stronger precipitation or runoff-based control on weathering rates than temperature (e.g. Oliva et al., 2003; Maher, 2010; Hayes et al., 2020), while global models (e.g. GEOCARB and derivatives) have often invoked temperature as a stronger control (Berner et al., 1983; Berner and Kothavala, 2001). Other global models, including GEOCLIM, have suggested that runoff may be a more significant control on weathering rates and thus long-term climate (Goddéris et al., 2014).

General Circulation Models (GCMs) have been used extensively to model climates on multi-million year timescales (e.g. Lunt et al., 2021). These models have been evaluated using records from proxies such as stable isotope ratios, TEX86, Mg\Ca ratios, and alkenones (e.g. Hutchinson et al., 2021). Although GCM simulations do not always produce results that agree with proxy data (e.g. Huber and Caballero, 2011; Keating-Bitonti et al., 2011; Lunt et al., 2012; Jagniecki et al., 2015), they have greatly expanded the spatial knowledge of past climates, whereas proxy data have been gathered from a comparatively
limited number of sites (e.g. Huber and Caballero, 2011). Furthermore, proxy data often produce conflicting reconstructions, such as differing atmospheric $CO_2$ levels (e.g. Jagniecki et al., 2015) or significantly different temperature and precipitation reconstructions (e.g. Keating-Bitonti et al., 2011). GCMs can be used to reconstruct global climates based on different proxy records and compare the results (e.g. Huber and Caballero, 2011; Lunt et al., 2012, 2016). Thus, GCMs have become a powerful tool in reconstructing palaeoclimates.

The large quantities of GCM-derived climate outputs have been incorporated into global geochemical models over the last 10-15 years (e.g. Berner and Kothavala, 2001; Donnadieu et al., 2004), enabling some of those models to provide spatially varying estimates of global weathering rates under different climate and palaeogeographic configurations. However, as noted above, these climate reconstructions can vary significantly depending on the model configuration and the boundary conditions used.



## 1.1 Controls on Mesozoic-Cenozoic Climate

Over multi-million year timescales, climates during the Mesozoic and Cenozoic were affected by a gradual increase in solar forcing and, in general, a gradual decrease in $CO_2$ forcing (Foster et al., 2017, ; Figure 1). A number of theories have been advanced to explain changes in atmospheric $CO_2$, and thus climate, over multi-million timescales. The gradual breakup of Pangaea and associated rifting likely resulted in increased volcanic degassing fluxes (Tajika, 1998). The late Cretaceous and the Cenozoic saw a period of mountain building associated with the latter stages of the break-up of Pangaea. The Laramide orogeny began approximately 70 Ma (Humphreys et al., 2003) and the development of the Northern Andes and Himalaya occurred during the early Cenozoic (Schellart, 2008). The development of new mountain ranges, particularly Himalaya, has been implicated in the reduction of $CO_2$ concentrations during the Cenozoic by increasing silicate weathering rates and thus increasing $CO_2$ drawdown (Raymo and Ruddiman, 1992). Changes in weathering fluxes have been linked to a number of climate shifts in the Earth's past, such as the global cooling which occurred during the Carboniferous (e.g. Goddéris et al., 2014, 2017).

Another direct forcing is palaeogeography, which can have substantial effects (Lunt et al., 2016). For example, the presence of mountain ranges may affect monsoon circulation and thus precipitation, or the presence of large continental interiors isolated from moisture can lead to large arid deserts. As such, the potential silicate weathering flux, or 'weatherability' of the continents is likely to represent a key climate forcing during the late Mesozoic-early Cenozoic and forms the focus of this study.

At steady-state, silicate weathering fluxes are equal to emissions from volcanic degassing (e.g. Walker et al., 1981; Berner et al., 1983; Goddéris et al., 2014). However, a distinction should be drawn between weathering (here, the chemical breakdown of silicate rocks by water and carbonic acid) and "weatherability". Weatherability corresponds to the susceptibility of the land surfaces to be weathered, under a given climate (temperature and runoff) (Kump and Arthur, 1997). Weatherability can be affected by the presence of land plants, or relief, or by changes in the lithology, or by the palaeogeographic configuration(Dessert et al., 2003; Oliva et al., 2003; West et al., 2005; Maher, 2010; West, 2012; Bazilevskaya et al., 2013).

Global palaeolithologies are poorly constrained and become even less so further back into the geological record. Similarly, factors such as topography and uplift rates, which affect erosion rates (e.g. Riebe et al., 2004; West et al., 2005; Gabet and Mudd, 2009), are also poorly constrained into geological time. In contrast, broad scale climate parameters like temperature and precipitation patterns are comparatively well constrained through proxy data and GCM studies. Climate states are in turn strongly controlled by the positioning of the continents, which can affect oceanic and atmospheric circulation patterns (e.g. Gyllenhaal et al., 1991; Von der Heydt and Dijkstra, 2006). The positioning of the continents also affects weatherability by determining the land area in intense weathering environments, such as the tropics (e.g. Goddéris et al., 2014).

In this study we focus on the impact of the palaeogeographic setting on the weatherability, assuming that all the other factors remain constant over the simulated period (this is certainly not the case, but our objective is to quantify exclusively the role of palaeogeography).



For an idealised case where degassing is constant (Walker et al., 1981), the global weathering flux will equilibrate the volcanic degassing of $CO_2$ and the global $CO_2$ consumption by silicate rock weathering will stay constant. But the evolving continental configuration modulates weatherability, allowing $CO_2$ to fluctuate (Goddéris et al., 2014).

In contrast to palaeo degassing rates, palaeogeographies are relatively well constrained. Thus, there is scope to investigate the impact of changing weatherability on long-term $CO_2$ concentrations. Early studies using zero-dimensional global models such as GEOCARB used estimations of global mean annual temperature (MAT) and runoff values, but no geographical constraints, to estimate global weathering fluxes and steady-state $CO_2$ concentrations in geological time (Berner, 1991). Further studies refined the GEOCARB approach by including palaeogeographical reconstructions, both conceptual (e.g. 'ringworld')

configurations) and realistic, which demonstrated that continental configurations had significant impact on steady-state $CO_2$ concentrations (e.g. Barron et al., 1989) and that palaeogeography and climate interact to determine steady-state $CO_2$ concentrations (Otto-Bliesner, 1995). Further model developments provided spatial patterns in climate forcing data, demonstrating that regional changes in weatherability can have global climate impacts (Donnadieu et al., 2004). More recently, such models have shown that the positioning of the continents also affects weatherability by determining the land area in intense weathering

environments, such as the tropics (e.g. Goddéris et al., 2014).

The impact of changing weatherability as a result of changes in palaeogeography during the Phanerozoic has previously been investigated by Goddéris et al. (2014) using the GEOCLIM model with climate inputs from the FOAM GCM, which found that the formation of supercontinents resulted in a continental configuration favourable for high atmospheric $CO_2$ concentrations (10-25 x PAL), as the converged continental configuration resulted in arid continental interiors and reduced weatherability.

Conversely, dispersed continental configurations were more favourable to lower atmospheric $CO_2$ concentrations (1-8 x PAL) by favouring higher continental runoff and weatherability. Furthermore, Goddéris et al. (2014) found that continents or even smaller landmasses crossing warm-humid climate belts can result in significant draw drown of atmospheric $CO_2$. For example, during the late-Triassic, the gradual northward drift of Pangaea brought larger areas of landmass into more humid zones. In response, $CO_2$ concentrations fell sharply over ~20 Myr from 19 x PAL to 3 x PAL (Goddéris et al., 2014).

Goddéris et al. (2014) had a relatively low temporal resolution of 22 simulations over the last ~520 Myr (approximately 1 simulation per 20-30 Myr) to produce a simulated $CO_2$ record, and from that glean insights on the effects of changing weatherability (through palaeogeographical changes) on long-term $CO_2$ concentrations and thus global climate. In contrast, this study will use 19 simulations over the Cretaceous-Eocene period (145-36 Ma) in addition to better constrained and higher resolution (factor of 3.65 increase) palaeogeography and climate inputs to produce our own simulated $CO_2$ record, assuming

a constant degassing rate. The higher temporal resolution will allow for the investigation of the impacts of higher temporal resolution changes (~5 Myr) in palaeogeography on long-term $CO_2$ and weatherability, while the higher spatial resolution will better constrain regional variability in weatherability relative to Goddéris et al. (2014). The use of climate inputs produced by the more complex HadCM3L model in this study (relative to the FOAM inputs used by Goddéris et al. (2014)) also represents an improvement. These model-based improvements should provide a more accurate picture of how global climate responds to



changes in weatherability, and better constrain the role of chemical weathering feedbacks on global climate through geological time.

## 2 Methodology

This study will investigate the impact of varying palaeogeography from the Early Cretaceous to the Late Eocene on weatherability, using a coupled global climate and geochemical model, GEOCLIM, with climatic reconstructions derived from the
HadCM3L model.

### 2.1 GEOCLIM

GEOCLIM is a coupled global climate and biogeochemical model initially developed in the early 2000's and has been used to investigate interactions between climate and geochemistry in deep time settings on geological timescales (Donnadieu et al., 2004). GEOCLIM uses temperature and runoff inputs from climate models at specified $CO_2$ concentrations (e.g. 280, 560 ppm)
and models the climate conditions through interpolation between as atmospheric $CO_2$ concentrations evolve within GEOCLIM simulations. GEOCLIM initially used CLIMBER inputs in early studies (Donnadieu et al., 2004), but more recent studies have used FOAM inputs (Lefebvre et al., 2013; Goddéris et al., 2014, 2017).

The COMBINE model within GEOCLIM handles biogeochemical processes for both land and ocean environments. Oceans are divided into 9 "boxes": 2 high latitude oceans (>60°N/S) (each separated into a photic layer and a deep ocean layer), a
low-mid latitude ocean (60°S - 60 °N) divided into photic, thermocline, and deep ocean layers, and an epicontinental sea also divided into photic and deep layers. A final 10th box represents the atmosphere. A range of biogeochemical processes are modelled within the ocean layers. As this study focuses on terrestrial processes we will omit a detailed description of the ocean processes, but descriptions can be found in Goddéris and Joachimski (2004).

Chemical weathering in GEOCLIM occurs in terrestrial grid cells, and the total sum value of silicate weathering from each
terrestrial grid cell, along with a prescribed volcanic degassing value is used to calculate changes in atmospheric $CO_2$ at each timestep. Chemical weathering calculations for silicate rocks are based on equations from Oliva et al. (2003) and Dessert et al. (2003).

$$F_{sil}(t) = k_{sil} * (\alpha_j(t) * \rho_j(t) * Ae^{-E_a/RT}) \tag{1}$$

Equation 1 calculates the granitic weathering flux in moles of C per year at a timestep *(t)* in a given grid cell, $F_{sil}(t)$, $k_{sil}$
is the silicate weathering constant, $\alpha_j(t)$ is the area of a grid cell ($10^6$ km$^2$), $\rho_j(t)$ is the runoff value of that grid cell (cm yr$^{-1}$). The final term (dimensionless) is an arrhenius equation for granite weathering, representing the granitic dependence on dissolution based on air temperature(T) and the universal gas constant (R), which uses the activation energy for granite ($E_a$) defined by Oliva et al. (2003) and the air temperature in a given grid cell, $T_j$.



$$F_{bas}(t) = k_{bas} * (\alpha_j(t) * \rho_j(t) * Ae^{-E_a/RT}) \tag{2}$$

A similar calculation exists for basalt (Equation 2) based on values in Dessert et al. (2003), and for carbonate rocks. Both calculations were derived from and calibrated on weathering reaction processes within granite and basaltic watersheds, described in detail in (Oliva et al., 2003) and (Dessert et al., 2003), respectively. We omit the carbonate equation here as this study focuses on variations in $CO_2$ drawdown via changes in silicate weathering fluxes. GEOCLIM assumes that each grid cell has an equal area of granitic, basaltic, and carbonate rocks. While this is obviously a significant simplification of the real-

world geology of the time, palaeolithologies are poorly constrained. In the absence of reliable global palaeolithologies from the studied time period, we assume an even distribution of lithologies for simplicity and consistency with Goddéris et al. (2014). A newer version of GEOCLIM, recently developed, includes an erosion dependency on chemical weathering rates based on equations in Gabet and Mudd (2009) (Maffre et al., 2021). As a key part of this study relates to comparing our new simulations to previous results with the GEOCLIM model (Goddéris et al., 2014) this version of the model was not used in this study for

reasons of consistency.

## 2.2   HadCM3L

HadCM3L is part of the HadCM3 "family" of coupled GCMs originally developed by the UK Met Office (Gordon et al., 2000; Valdes et al., 2017). The HadCM3 family of models have been in use for over 15 years, with various modifications to the configuration of the original HadCM3 model being made in that time. HadCM3L is a version of HadCM3 using a

"low-resolution" ocean, where ocean and atmosphere share the same 96x73 (3.75° x 2.5°) resolution (Valdes et al., 2017). Although described as "low-resolution", 96x73 is nonetheless 3.65 times higher resolution than FOAM. However, like FOAM, HadCM3L is used for long term simulations where higher resolution models would be too time intensive to be practical. To that end, HadCM3L has been used extensively in pre-quaternary climate studies, where long term climate simulations are required (e.g. Lunt et al., 2010, 2011, 2012; Valdes et al., 2017). This study uses climate inputs from 19 HadCM3L simulations of the

Cretaceous-Eocene period at intervals of 3-13 Ma, the same simulations used by Farnsworth et al. (2019). These simulations were run with $CO_2$ concentrations at 560 and 1120 ppm, which considered to be a reasonable representation of potential $CO_2$ concentrations during the Creataceous-Eocene, which are subject to considerable uncertainty (Lunt et al., 2016). The simulations in Farnsworth et al. (2019) (used for this study) differ from those in Lunt et al. (2016) slightly in that they have a spin up time of 10,422 years, as opposed to 1422 years in Lunt et al. (2016). The extended spin up time was to enable the

simulations to more closely approach an equilibrium state and to make use of the fully dynamic mode within the coupled vegetation model. Runoff calculations are affected by the vegetation model, with both canopy interactions and soil infiltration considered within HadCM3L. A full description of the runoff parameterisation can be found in Gregory et al. (1994).

The palaeogeographic reconstructions used in this study were developed by Getech for Lunt et al. (2016). The reconstructions were produced using methods developed by Markwick and Valdes (2004), and include data from well-constrained geological

databases such as The Paleogeographic Atlas Project. These palaeographries were originally constructed at 0.5$^o$ x 0.5$^o$ resolu-





tion - from these high resolution palaeogeographies topographies, bathymetries, and land-sea masks were generated at $3.75^o$ x $2.5^o$ resolution for the HadCM3L model (Lunt et al., 2016). Additionally, HadCM3L is coupled to the TRIFFID model (Top-down Representation of Interactive Foliage and Flora Including Dynamics), which is a dynamic global vegetation model (Cox et al., 2002). The two modules are coupled via the MOSES 2.1 land surface scheme (Cox et al., 1999). The TRIFFID model calculates the fraction of five plant functional types (broadleaf trees, needleleaf trees, $C_3$ grasses, $C_4$ grasses, and shrubs) in each grid cell. Although these are modern plant types, studies have suggested that models such as TRIFFID can nonetheless perform acceptably in providing vegetation feedback signals back to at least 250 million years (Donnadieu et al., 2009; Zhou et al., 2012).

### 2.3 GEOCLIM Configuration

The degassing rate for the paleogeography simulations was set to a constant value of 1 x $10^{13}$ mol C yr$^{-1}$, similar to previous GEOCLIM studies (Lefebvre et al., 2013). A value of 1 x $10^{13}$ mol C yr$^{-1}$ is within the range of estimates (˜1.2 – 2 times modern) for degassing rates during the Cretaceous-Eocene period (Van Der Meer et al., 2014). These simulations were also run using variable degassing rates ranging from a low value of 7.82 x $10^{12}$ mol C yr$^{-1}$ in the latest-Eocene to a high of 1.23 x $10^{13}$ mol C yr$^{-1}$ in the early-Cretaceous (Appendix Figure A1). This range is based on values calculated by Van Der Meer et al. (2014) using reconstructed subduction zone arc lengths.

We conducted two sets of experiments. We run one simulation per time slice, fixing the atmospheric $CO_2$ to 2.85 times the pre-industrial value of 280 ppm (˜ 800 ppm). This $CO_2$ concentration is intermediate between the 560 and 1120ppm reconstructions. The first set of experiments allows us to isolate the effect of the palaeogeography on the weatherability of the continental surfaces, all other factors being fixed. The second set of experiments was conducted until atmospheric $CO_2$ reaches steady-state. For each of those steady states, the total $CO_2$ consumption by continental silicate rock weathering balances the solid Earth degassing.

### 3 Results

Our 19 simulations produced a simulated $CO_2$ record from 145-36 Ma. Throughout the results and discussion, patterns within the record will be compared with changes in global weathering fluxes and climate variables to investigate if, and how, changes in weatherability have affected the $CO_2$ record. To begin, we discuss the role of degassing and silicate weathering fluxes in driving these variations.

The GEOCLIM simulations of the 19 time slices resulted in a range of $CO_2$ concentrations, with the lowest value of ˜650 ppm at the end of the Eocene and the highest value of ˜1100 ppm during the early Cretaceous (Figure 1). Atmospheric $CO_2$ exceeds 1000 ppm during the early Cretaceous but declines significantly to ˜700 ppm by the mid-Cretaceous. There is a brief increase in $CO_2$ around 85 Ma to ˜900 ppm, followed by a gradual decrease into the Cenozoic and through to the end of the Eocene. The simulations run with variable degassing rates produced a similar pattern, albeit with higher atmospheric $CO_2$





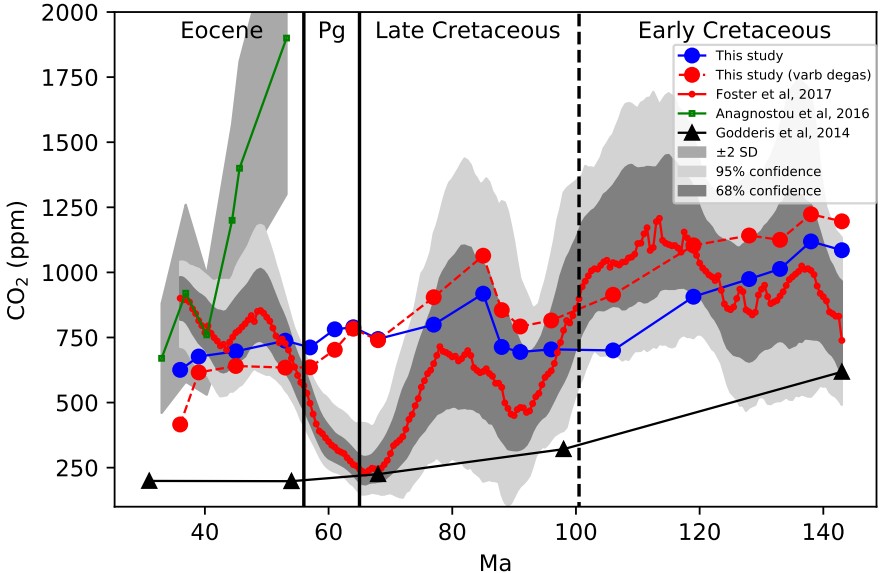

**Figure 1.** Steady-state $CO_2$ as modelled by GEOCLIM plotted against two $CO_2$ proxy records from the literature and simulated record from Goddéris et al. (2014). The GEOCLIM model output displays some agreement with the record from Foster et al. (2017), particularly during the early-Cretaceous where $CO_2$ concentrations are comparably high, the late-Cretaceous, and during the late-Eocene. The record from Anagnostou et al. (2016) displays considerably higher $CO_2$ concentrations during the early-Eocene than those in the GEOCLIM and Foster records, but shows better agreement with both records towards the mid and late Eocene. A second GEOCLIM model output (dashed red) displays the results of running GEOCLIM simulations using variable degassing rates calculated in Van Der Meer et al. (2014).

concentrations during the Early Cretaceous and lower concentrations in the Late Eocene relative to the simulations using a constant degassing rate.

Exploring now the simulations performed with a fixed atmospheric $CO_2$ level at 2.85 times the pre-industrial value, the calculated global silicate weathering flux is inversely related to steady-state $CO_2$ (Figure 2). The early Cretaceous is marked by low silicate weathering fluxes, although fluxes increase significantly towards the mid Cretaceous and peak at approximately 91 Myr. A brief, but sharp decrease occurs until 85 Myr, followed by a similarly rapid recovery at the end Cretaceous to a similar level of weathering seen at 91 Myr. Another brief decrease occurs into the Palaeocene, followed by a gradual rise through the Eocene. Initial global mean temperature varies little throughout the Cretaceous-Eocene period, although a period

of warming occurs in the mid-late Cretaceous, followed by gradual cooling into the Cenozoic. Farnsworth et al. (2019) noted in their simulations that temperature varied little with palaeogeography. Initial mean global runoff increases significantly into the mid-Cretaceous, followed by a brief decrease around 85 Ma, and then an increase again towards the end-Cretaceous. Runoff drops sharply at the start of the Palaeocene and remains mostly stable through the Eocene.





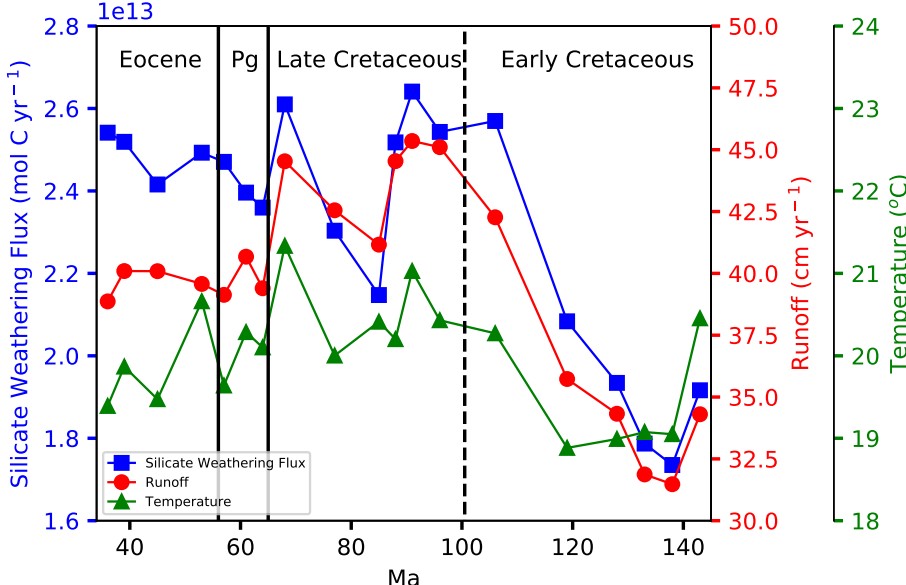

**Figure 2.** Modelled silicate weathering fluxes (blue squares) plotted against global mean annual runoff (red circles) and temperature (green triangles). Trends in silicate weathering fluxes are similar to those of global runoff, but show less similarity to temperature.

In the fixed $CO_2$ simulations, the distribution of runoff changes significantly during the Cretaceous-Eocene period (Figure 3). During the early Cretaceous, large areas of North America, South America, Asia, and Antarctica have low or zero runoff, likely due to low moisture transport from the oceans given the converged continental configuration at this time. There are some areas of high runoff, such as the northern coast of Gondwana, along with parts of the western Amazon and east Africa. Into the mid-late Cretaceous, North America and Antarctica become much more humid while Asia becomes more arid. The Amazon remains persistently humid, while the Atlantic coast of North America has high runoff, possibly due to increases in evaporation (and or tropical cyclones) associated with the expansion of the Atlantic. The west coast of North America has especially high runoff, with zonal means indicating that runoff here is greater than at the equator, although runoff falls towards the late Cretaceous. Into the Cenozoic, areas such as India and western Australia have a significant increase in runoff. India crosses the equator at this time, so the increased runoff may be related to interactions with the ITCZ. Towards the end of the Eocene, areas such as North America and Asia become more arid, while runoff in the southern mid-latitudes increases slightly. Much of the world however becomes more arid, reflecting the global drop in runoff at the end of the Eocene, possibly linked to the general cooling trend and falling evaporation rates from the oceans from around 49 Ma into the Eocene-Oligocene boundary.





**Figure 3.** Regional mean annual runoff (cm yr$^{-1}$ maps from the earliest-Cretaceous (s) to the latest-Eocene (A). An arid climate prevails during the early-Cretaceous (s-o), particularly in the continental interior of North America. Runoff increases significantly from the mid-Cretaceous, with very high runoff rates present in Amazonia (m-k). High runoff totals are present in India during the latest-Cretaceous and the Palaeocene as it crosses the equator (h-e). The world becomes slightly more arid through the Eocene (d-a).

## 4 Discussion

Silicate weathering fluxes at 2.85 the pre-industrial $CO_2$ show a strong relationship with mean global runoff rates and a strong
inverse relationship with atmospheric $CO_2$ concentrations (Figures 1 and 2, r = 0.88 and -0.96, respectively), indicating that





changes in global runoff are the primary control on steady-state $CO_2$ concentrations over the modelled period. However, there are some inconsistencies in these trends. Notably, global silicate weathering fluxes increase towards the late Eocene despite a fall in global runoff rates. To understand these inconsistencies in the global trend, we investigate the relationship between runoff rates and silicate weathering fluxes at the regional level.

**4.1 Regional Trends**

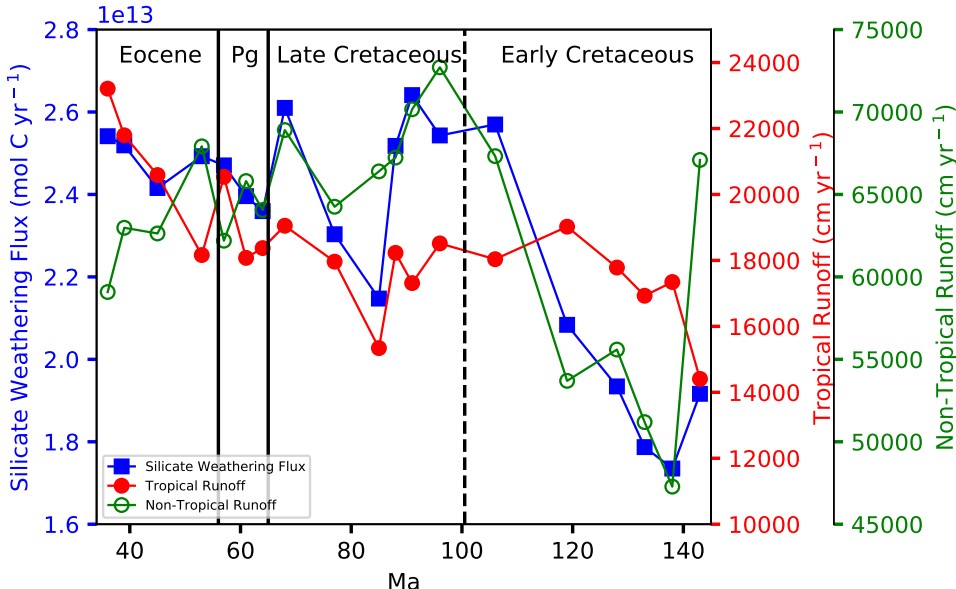

**Figure 4.** Modelled silicate weathering fluxes (blue squares) plotted against total tropical runoff (red circles) and total non-tropical runoff (green open circles). The tropics are defined here as all areas within 30°N/S. Changes in silicate weathering fluxes are driven broadly by non-tropical runoff during the early-Cretaceous and by tropical runoff during the Cenozoic.

During the Eocene, both mean and total global runoff from the fixed $CO_2$ runs falls but silicate weathering fluxes rise, an inverse of the pattern seen during the Cretaceous where runoff rates and weathering fluxes show identical patterns (Figure 4). During the early-Cretaceous, silicate weathering fluxes are strongly controlled by non-tropical runoff (>30°N/S), showing an almost identical pattern until the mid-Cretaceous (Figure 4). This pattern weakens somewhat around 86 Ma, but strengthens 245 by the end of the Cretaceous. Into the Cenozoic, the pattern weakens again and silicate weathering fluxes become anti-phased to non-tropical runoff. In contrast, silicate weathering fluxes in the Eocene appear to be more strongly controlled by runoff rates in the tropics (Figure 4). Similarly, tropical runoff changes in the Cretaceous does not appear to have a significant impact on silicate weathering fluxes, with the exception of a brief period during the late-Cretaceous (86 Ma). There is a noticeable





weakening of the influence of non-tropical runoff on weathering fluxes from the Cretaceous to the Cenozoic, although due to
the small number of simulations in this period (n = 6), confidence in this relationship is low.

The shift from non-tropical to tropical runoff controlled weathering appears to represent a regime change in the long-term
climate pattern seen during the modelled period. Theoretically, a continental configuration with a greater land area in the
low-latitudes, where precipitation is generally highest, would favour lower global $CO_2$ concentrations by increasing silicate
weathering rates (Gibbs and Kump, 1994; Otto-Bliesner, 1995; Goddéris et al., 2014). However, changes in continental posi-
tioning will alter ocean circulation and evaporation patterns, which would then alter climate patterns, especially runoff (Barron
et al., 1989; Lunt et al., 2012). Furthermore, runoff patterns will also be affected by whether continents are in a dispersed or
converged (i.e. supercontinental) configuration.

The shift in weathering controls seen in the GEOCLIM simulations may be the result of a change in the effects of palaeo-
geography on weathering fluxes. During the Cretaceous, the models suggest that increased weathering occurred as a result
of the climate becoming more humid due to the break-up of Pangaea. The break-up of Pangaea promoted higher weathering
fluxes through evaporation by increased ocean area and more favourable moisture transport to the continental interiors. In
contrast, during the Cenozoic both total land areas and tropical land areas increased. During the same period, modelled total
global runoff falls, but tropical runoff increases, resulting in both a greater weatherable area and a more intense weathering
environment in the tropics. As such, the shift in weathering controls in the Cenozoic from non-tropical to tropical runoff may
be the result of the continents moving into a configuration which promotes higher weatherability, while during the Cretaceous
changes in weathering fluxes were largely caused by climate changes induced by the break-up of Pangaea which increased
runoff in the previously arid continental interiors.

### 4.2 Comparison to Proxy Data

While the GEOCLIM simulations in this study represent an improvement over previous studies due to their higher spatial-
temporal resolution and better constrained palaeogeography, they are nonetheless an obvious simplification relative to both
pure GCM studies (which provide more realistic modelling of climate change processes, and feedbacks in particular, rather
than the linear interpolation used here) and real-world settings. Because the primary aim of this study is to assess the impact of
palaeogeographic changes on potential global "weatherability", simplifications such as a uniform palaeolithologies were used
in the absence of well-constrained data for such fields. Still, it is naturally of interest to compare the $CO_2$ record produced in
this study with those derived from previous modelling studies and proxy data. Such a comparison may provide an indication
of the potential impact of changing weatherability on long-term $CO_2$ concentrations through the Cretaceous-Eocene period.

Figure 1 presents the $CO_2$ output produced in this study plotted (henceforth referred to as the GEOCLIM model output)
against two $CO_2$ records derived from proxy data (Anagnostou et al., 2016; Foster et al., 2017) and the 145-34 Ma portion
of the record from Goddéris et al. (2014). Through the Cretaceous-Eocene period, there are three periods in which the trends
in the GEOCLIM model output agrees with the two proxy records: the early-Cretaceous, the late Cretaceous, and the mid-
Eocene. Both the GEOCLIM model output and the record from Foster et al. (2017) indicates relatively high (>800 ppm) $CO_2$



concentrations in the earliest-Cretaceous, followed by a general decreasing trend until 125 Ma. The high $CO_2$ concentrations in the early-Cretaceous are likely the result of the converged continental configuration which is unfavourable for high weathering fluxes. Both records indicate a rise in $CO_2$ just after the earliest-Cretaceous (138 Ma, although only one GEOCLIM simulation is available between 140 and 135 Ma) . At this time, the continental interiors become more arid, as reflected by global runoff falling from ~82 x $10^3$ cm $yr^{-1}$ to ~64 x $10^3$ cm $yr^{-1}$ over a period of 5 Myr (Figures 2 and 3). The fall in global runoff leads to a reduction of silicate weathering fluxes, resulting in a $CO_2$ increase.

A rise in $CO_2$ during the late-Cretaceous (~85 Ma) occurs in both the GEOCLIM model output and the record from Foster et al. (2017). During the same period, a drop in global weathering fluxes (especially in the tropics) and a fall in total tropical runoff occurs (Figure 4). In both records, the rise in $CO_2$ is relatively brief (10-20 Myr) and of approximately the same magnitude (~200 ppm).

All three records indicate a drop in $CO_2$ of varying magnitudes towards the mid-Eocene, with the GEOCLIM model output showing the smallest drop in $CO_2$ ($< 100$ ppm) while the record from Anagnostou et al. (2016) indicates a much larger drop of around 600 ppm. A fall in $CO_2$ from the early to mid-Eocene is often attributed to increased silicate weathering (e.g. Raymo and Ruddiman, 1992). Indeed, our modelled Eocene weathering fluxes are generally high and with a general rising trend towards the end of the Eocene, which coincides with falling $CO_2$ concentrations (Figures 1 and 2). Zonal weathering plots indicate a wider area of high weathering fluxes in the tropics (focused on Eastern Asia and India) at this time relative to other time periods (e.g. mid-Cretaceous) in this study ($30°$ N/S)(Appendix Figure A2).

### 4.2.1 Impact of Variable Degassing Rate

Based on calculations from Van Der Meer et al. (2014), degassing rates peak in the early-Cretaceous and fall gradually through the late-Cretaceous and Cenozoic periods. The variable degassing simulations produced a record similar to the GEOCLIM simulations using a fixed degassing rate of 1 x $10^{13}$ mol C $yr^{-1}$, which is similar to the range of degassing rates in Van Der Meer et al. (2014). In general, the variable degassing simulations produced slightly higher steady-state $CO_2$ concentrations during the Cretaceous, but slightly lower steady-state concentrations during the Cenozoic (Figure 1). Steady-state $CO_2$ concentrations fall sharply in the latest-Eocene in response to a decrease in degassing rates that continues into the Neogene. The similarity of both the constant and variable degassing $CO_2$ timeseries implies that on timescales of tens of millions of years, and in the context of this time period and the weathering model used, changes in weathering have a greater control on $CO_2$ than changes in degassing.

### 4.3 Comparison to Previous GEOCLIM Palaeogeography Study

### 4.3.1 Comparison of Results With Previous Work

The model outputs presented in this study demonstrate that the long-term climate (under constant degassing rates) is strongly controlled by silicate weathering fluxes, which in turn are predominately controlled by runoff rates. Palaeogeographical changes are likely to have had significant impacts on both the intensity and distribution of runoff. These findings are in line with those





of Goddéris et al. (2014) , which found a strong direct palaeogeographical influence on runoff, and thus weathering rates
and long-term atmospheric $CO_2$. Both Goddéris et al. (2014) and this study found a general decreasing trend in atmospheric
$CO_2$ from the Cretaceous to the early Cenozoic, although this study has a higher temporal resolution during that period (19
simulations vs 5). Goddéris et al. (2014) noted a number of potential climate impacts from palaeogeographical changes, based
on their GEOCLIM study with FOAM inputs.

Goddéris et al. (2014) found that a converged, or supercontinental, arrangement inhibited weathering fluxes leading to high
$CO_2$ concentrations. In contrast, a dispersed continental configuration favours higher weathering fluxes. While the time period
in this study does not cover the formation of Pangaea, during the earliest Cretaceous the continents were still in a converged
configuration (Figure 3) and coincided with the highest $CO_2$ concentrations and lowest weatherability in the modelled period.
Goddéris et al. (2014) showed significant rises and falls in $CO_2$ (up to 20x PAL) within ~10 Myr occurring during the formation
and subsequent break-up of Pangaea associated with changes in weathering fluxes. Similarly, this study indicates rapid changes
in $CO_2$ (200-300 ppm within 5 Myr) as Gondwana and Laurasia break apart, indicating the potential for geologically rapid
$CO_2$ changes as a result of palaeogeographical changes.

Goddéris et al. (2014) also noted the potential climate impact of high weathering rates on small continental landmasses in
tropical areas. During the Rhaetian period of the Triassic, south China crossed the tropics and contributed 17% of the global
$CO_2$ drawdown despite a small land area relative to Pangaea (Goddéris et al., 2014). This disproportionate $CO_2$ drawdown
may be the result of a generally arid climatic context during the Triassic, as a similar result is not seen in this study (Figure
3), despite the passage of India across the tropics during the Cretaceous-Eocene. Although runoff rates (and thus weathering
rates) on the Indian subcontinent increase significantly as it crosses the tropics, its passage does not drive a large decrease in
atmospheric $CO_2$ (Figures 2 and 3). The rise in weathering rates occurs just after a global drop in runoff rates and weathering
fluxes. Thus, while the passage of India through the tropics raises global weathering fluxes significantly, the effect of this rise is
counteracted by the fall in global weathering fluxes as a result of increasing aridity. As such, the influence of small continental
landmasses on long-term $CO_2$ concentrations appears to be variable and largely dependent on the prevailing global climate
state, with $CO_2$ concentrations more sensitive to change under a more arid global climate. In this study, a stronger influence
on long-term weathering rates occurs as the result of a continental configuration that favours either increased evaporation from
the oceans and moisture transport to continental interiors or a general increase in tropical land areas.

**4.3.2  Evaluation of the Role of GCM**

A notable advance made by this study on the work of Goddéris et al. (2014) is the higher resolution GCM and palaeogeography
inputs, significantly improving our ability to asses the impact of regional changes in climate and palaeogeography on global
weathering fluxes. A comparison between a FOAM and a HadCM3L simulation of the early Eocene (52Ma) reveals significant
differences between the two inputs used and produced substantial differences in steady-state $CO_2$ (299 ppm and 505 ppm,
respectively). Furthermore, a step-by-step transition of variables from all FOAM to all HadCM3L inputs was also undertaken,
to investigate the independent impact of each of the differences between the two GCMs. Initially, the FOAM climate variables





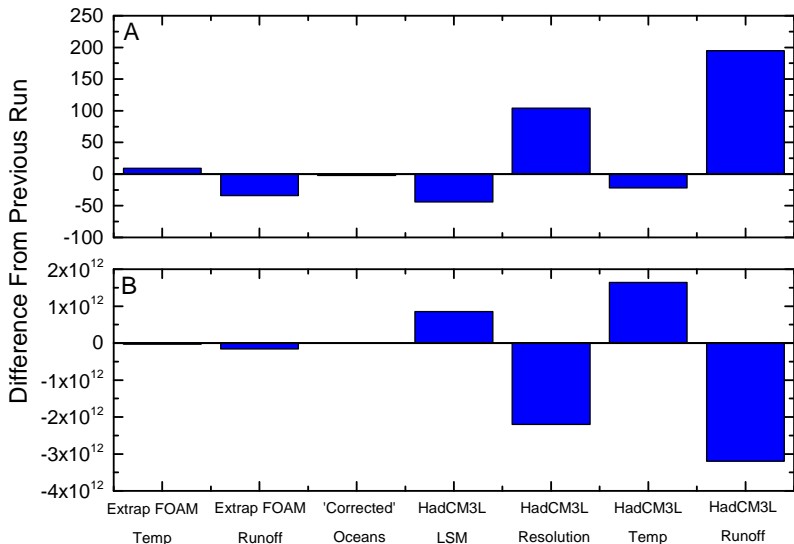

**Figure 5.** Impact of incremental changes between FOAM and HadCM3L climate inputs in GEOCLIM (A) Changes in modelled steady-state $CO_2$ in ppm (after 1 Myr) for input changes relative to the previous run and (B) changes in global silicate weathering fluxes at $CO_2$ fixed to 2.85 times the pre-industrial level (mol C $yr^{-1}$) relative to the previous run. The variable changed in each run is shown as the x-axis. Changes in weathering fluxes are essentially an inverse of the pattern of changes in steady-state $CO_2$.

were extrapolated using the same method used to generate climate data at 11 $CO_2$ concentrations for the HadCM3L data (Appendix Equations A.1 and A.2). This process allowed us to test whether our interpolation resulted in any significant deviation from the original dataset. Then a minor change to the ocean layout was made, followed by the introduction of a low-resolution 350 (48x40) version of the HadCM3L LSM. Then the model resolution was increased to match the HadCM3L resolution (96x73). The HadCM3L temperature data was then included, replacing the FOAM temperature inputs, followed finally by replacing the FOAM runoff inputs with those from the HadCM3L simulation.

The step-wise transition between FOAM and HadCM3L revealed that the majority of the differences in steady-state $CO_2$ concentrations can be explained by differences in model resolution and runoff (Figure 5), as these produced the greatest changes 355 in steady-state $CO_2$ values. HadCM3L produces a globally more arid world than the FOAM simulation and shows far greater regional variability. These results are significant as previous studies (both from field and model data) have suggested that a significant proportion of global weathering fluxes may be the result of so-called "hot-spots", i.e. small land areas with high weathering fluxes such as volcanic islands and mountain ranges (e.g. Kent and Muttoni, 2013). While FOAM inputs show areas of higher runoff associated with mountain ranges, the higher resolution of the HadCM3L inputs relative to FOAM inputs may 360 be sufficient to more accurately resolve weathering hot spots. Furthermore, given that weathering fluxes are highly spatially variable, the HadCM3L simulation likely better constrains such features and thus provides a better estimate of global weather-



ing fluxes. This study indicates a number of areas of high weathering fluxes, particularly outside of the tropics, associated with areas of high relief in the GCM simulations, particularly the Laramides and Appalachians (north America) and the southern Andes (Figure 3). In the original HadCM3L simulations these areas have high runoff rates, likely associated with orographic intensification of rainfall. Thus, the GEOCLIM simulations in this study may provide a better spatial estimate of weathering fluxes than those using FOAM inputs. These regions may be partly responsible for the sharp rise in non-tropical runoff in the early-Cretaceous (Figure 4), and further support the role of active mountain ranges in influencing $CO_2$ drawdown (Raymo and Ruddiman, 1992; Riebe et al., 2004; West et al., 2005; Goddéris et al., 2017).

### 4.4 Implications for Palaeoclimates

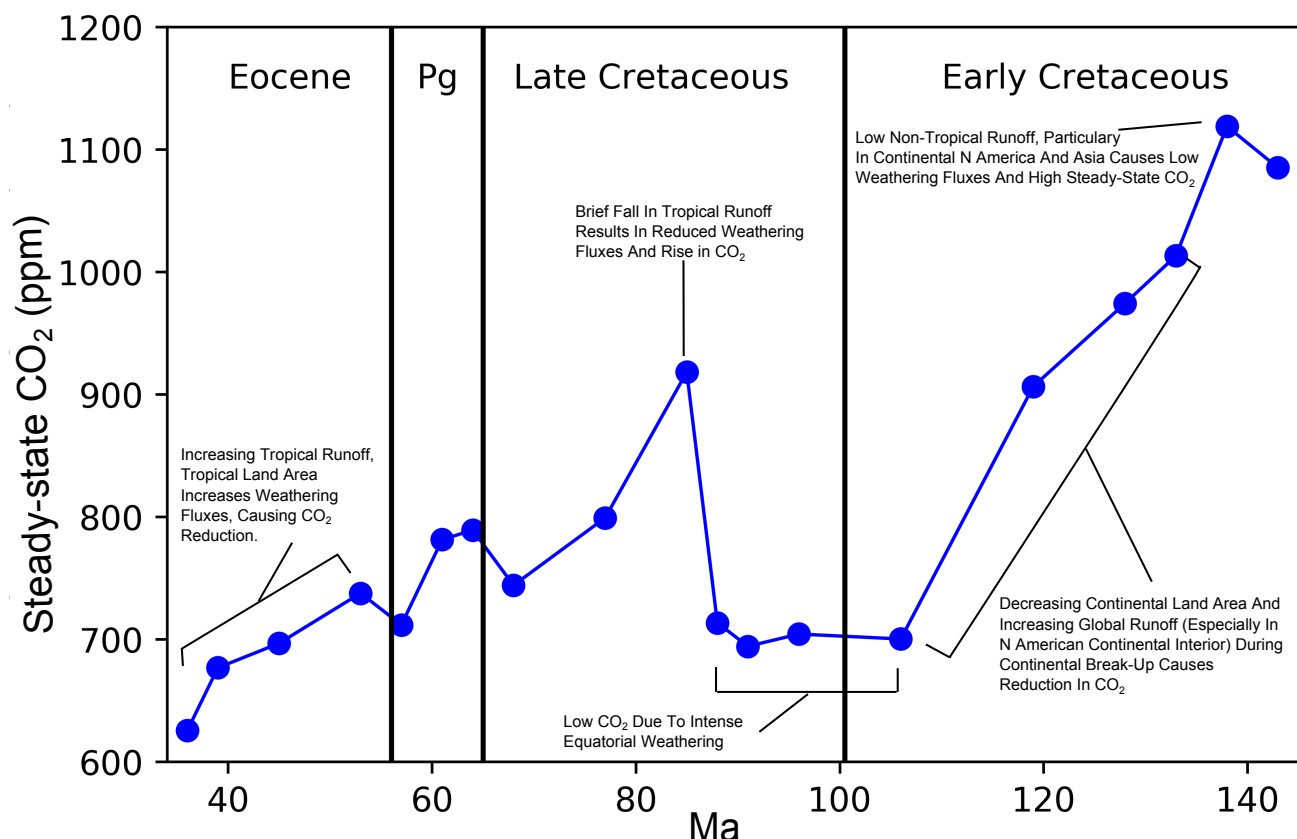

**Figure 6.** GEOCLIM steady-state $CO_2$ record with annotations showing major controls on steady-state $CO_2$ concentrations. Significant changes in steady-state $CO_2$ are typically due to changes in runoff and continental land areas.

The GEOCLIM $CO_2$ record presented in this study reflects the changes in global "weatherability" through the Cretaceous-Eocene period, indicating a general trend towards increased global weatherability during that time, punctuated by a decrease



at ~80 Ma. Although comparison with $CO_2$ records from proxy data suggest that not all changes in long-term $CO_2$ can be ascribed to changes in global weatherability, the three records indicate that changes in global weatherability have sufficiently powerful impacts on geologically short timescales (~10 Myr) to be expressed in long term $CO_2$ records (i.e. hundreds of ppmv),

further supported by periods of agreement between the records. These changes in weatherability are primarily associated with the break-up of Pangaea during the early-Cretaceous, and changes in tropical land areas and runoff rates in the mid-Cretaceous and the Eocene, respectively. Based on these findings, the implications for palaeoclimates are twofold.

Firstly, this study reinforces the conclusions of previous studies that continental configuration is a strong control on long-term $CO_2$ concentrations by influencing runoff (Otto-Bliesner, 1995; Gibbs et al., 1999; Donnadieu et al., 2004; Goddéris et al.,

2014). Supercontinental configurations reduce global weatherability and favour high $CO_2$ concentrations by limiting global runoff, while dispersed configurations increase global weatherability and favour lower $CO_2$ concentrations as the dispersed configuration favours moisture transport to continental interiors.

Secondly, this study demonstrates the potential for relatively localised changes in weatherability to have global impacts. Such local changes may result in a temporary decoupling of globally averaged runoff rates from globally averaged weathering rates.

During the Eocene, non-tropical runoff totals fall substantially but tropical runoff increases significantly, likely associated with an increase in tropical land areas at the same time. Similarly, during the mid-Cretaceous, tropical runoff totals remain stable but non-tropical rainfall increases. The increase in non-tropical rainfall is sufficient to increase silicate weathering fluxes and lower atmospheric $CO_2$ concentrations by 200 ppm over 13 Myr.

## 5 Conclusions

This study explores the potential impact of changing "weatherability" during the Cretaceous-Eocene period on long-term $CO_2$ concentrations, as well as the impact of using high-resolution GCM data on modelled weathering rates and steady-state $CO_2$ concentrations. The palaeogeography analysis showed that "weatherability" changed significantly through the Cretaceous-Eocene period as Pangaea broke up and the continents entered a more dispersed configuration. Changes in weatherability as a result of palaeogeographical change are due to two processes. The first process is increased evaporation and more favourable

moisture transport as continents became more dispersed, which occurred during the early-Cretaceous. The second process is due to continents moving into more humid zones, which occurred during the late-Eocene.

Steady-state $CO_2$ concentrations were initially high in the early-Cretaceous due to low total global runoff inhibiting weathering fluxes. As Pangaea broke up, evaporation from the ocean increased and improved moisture transport to the continental interiors, increasing runoff rates and weathering fluxes, resulting in lower steady-state $CO_2$ concentrations. Into the Cenozoic

however, global weatherability appears to "switch" regimes. In the Cenozoic, weatherability appears to be determined by increases in tropical land area, allowing for greater weathering in the tropics. Furthermore, global runoff fell in the late Eocene but silicate weathering fluxes continued to increase. The increase in silicate weathering is due to an increase in total tropical





runoff as land areas in the tropics increased, which is sufficient to allow global weathering fluxes to increase despite a fall in total global runoff.

We also investigated and quantified the impact of using different GCM datasets in GEOCLIM. The high-resolution HadCM3L datasets produced substantially different steady state $CO_2$ concentrations relative to FOAM datasets, primarily due to changes in runoff reconstructions. The use of high-resolution GCM data was instrumental in revealing these potential impacts, and highlights the benefits of such data in palaeo weathering studies.

    While the aim of this study was to investigate the role of palaeogeography on weatherability, the $CO_2$ record produced by
GEOCLIM was also compared with two proxy derived records from the literature to determine what aspects, if any, of long-term $CO_2$ change could have been driven by changes in weatherability. Three periods were identified when the GEOCLIM model output agreed with the proxy $CO_2$ records and thus suggest that changes in weatherability were perhaps influencing global climate: the early-Cretaceous, the late-Cretaceous, and the mid-Eocene. High $CO_2$ concentrations in the early Cretaceous are likely due to the arid nature of the continental interior. A brief (10-20 Myr) rise in $CO_2$ during the late-Cretaceous
appears to be the result of falling tropical runoff, causing global weathering fluxes to fall. Finally, a falling trend in $CO_2$ during the mid-Eocene appears to be linked to increased global weatherability as tropical land areas increase.

    The periods of agreement in the three $CO_2$ records indicate that weatherability changes were sufficient in magnitude to affect the long-term climate of the Cretaceous-Eocene period. Furthermore, this study shows the potential for regional changes (e.g. localised changes in tropical runoff and land area) to have global impacts, and that such regional changes may be more
significant than global averages for determining long-term climate.

*Author contributions.* Nick Hayes: Conceptualization, Methodology, Formal Analysis, Investigation, Writing – Original Draft

    Daniel Lunt: Conceptualization, Methodology, Resources, Writing – Review and Editing, Supervision

    Yves Goddéris: Methodology, Resources, Writing – Review and Editing

    Richard Pancost: Conceptualization, Writing – Review and Editing, Project Administration, Funding Acquisition

Heather Buss: Writing – Review and Editing, Supervision, Project Administration

*Acknowledgements.* We thank Yannick Donnadieu (Aix Marsaille University) for providing FOAM simulation data and providing helpful comments during this project, as well as those who have provided constructive feedback on the manuscript. This work was supported by European Research Council under the European Union's Seventh Framework Programme (FP/2007-2013)/ERC Project "The Greenhouse Earth System" (T-GRES), Grant Agreement no. 340923.
The views expressed in the paper are those of the authors and do not necessarily reflect the position of the Environment Agency.





*Competing interests.* Some authors are members of the editorial board of Climates of the Past.



**References**



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

**Appendix A**

Because GEOCLIM interpolates climate variables based on climate data at specified $CO_2$ concentrations, it was necessary to ensure that both the FOAM and HadCM3L input data had consistent $CO_2$ concentrations. FOAM was previously run at 11 $CO_2$ concentrations between 160 and 1400 ppm, while HadCM3L was previously run only at 560 and 1120 ppm. For consistency, a methodology was developed to produce "synthetic" model output from HadCM3L at the same $CO_2$ concentrations as for

FOAM. This was deemed particularly important because GEOCLIM model runs with Ypresian FOAM inputs have previously produced $CO_2$ concentrations below 560 ppm (Lefebvre et al., 2013). In GEOCLIM, if modelled $CO_2$ concentrations exceed the available $CO_2$ range, the climate variables will cease to update.

To address these issues, the HadCM3L climate inputs were extrapolated to the same 11 $CO_2$ concentrations to cover the same range as the FOAM data. Two separate extrapolation methods were used to calculate the temperature and runoff to better reflect

their sensitivity to $CO_2$ concentration changes in observed data. The HadCM3L temperature $(T_x)$ for a $CO_2$ concentration of $CO_{2x}$, was approximated as:

$$T_x = T_{560} + (T_{diff} * \frac{ln\frac{CO_{2x}}{280}}{ln2} - 1) \tag{A1}$$

Temperature values were extrapolated based on absolute differences between the 560 and 1120 ppm temperature outputs

(Equation A.1). $T_x$ is the temperature value at any given $CO_2$ concentration, where $T_{560}$ is the temperature at 560 ppm, $T_{diff}$ is the difference in temperature between 1120 and 560 ppm, and $CO_{2x}$ is a specific concentration of atmospheric $CO_2$ (ppm).

Runoff was extrapolated in a similar fashion to temperature, except that a constant ratio rather than a constant absolute difference was assumed per $CO_2$ doubling, because this relationship is found in the model results and absolute value changes would result in large areas of negative runoff and excessive aridity at lower $CO_2$ concentrations which were deemed to be

unrealistic:



$$R_x = R_{560} * exp((\frac{CO_{2x} - 560}{560} * ln(R_{diff}))$$ (A2)

Equation A.2 was used to extrapolate runoff rates (cm yr$^{-1}$), $R_x$, at any given CO$_2$ concentration. $R_{560}$ represents the runoff rate at 560 ppm, while $R_{diff}$ is the magnitude difference between runoff values at 1120 and 560 ppm.

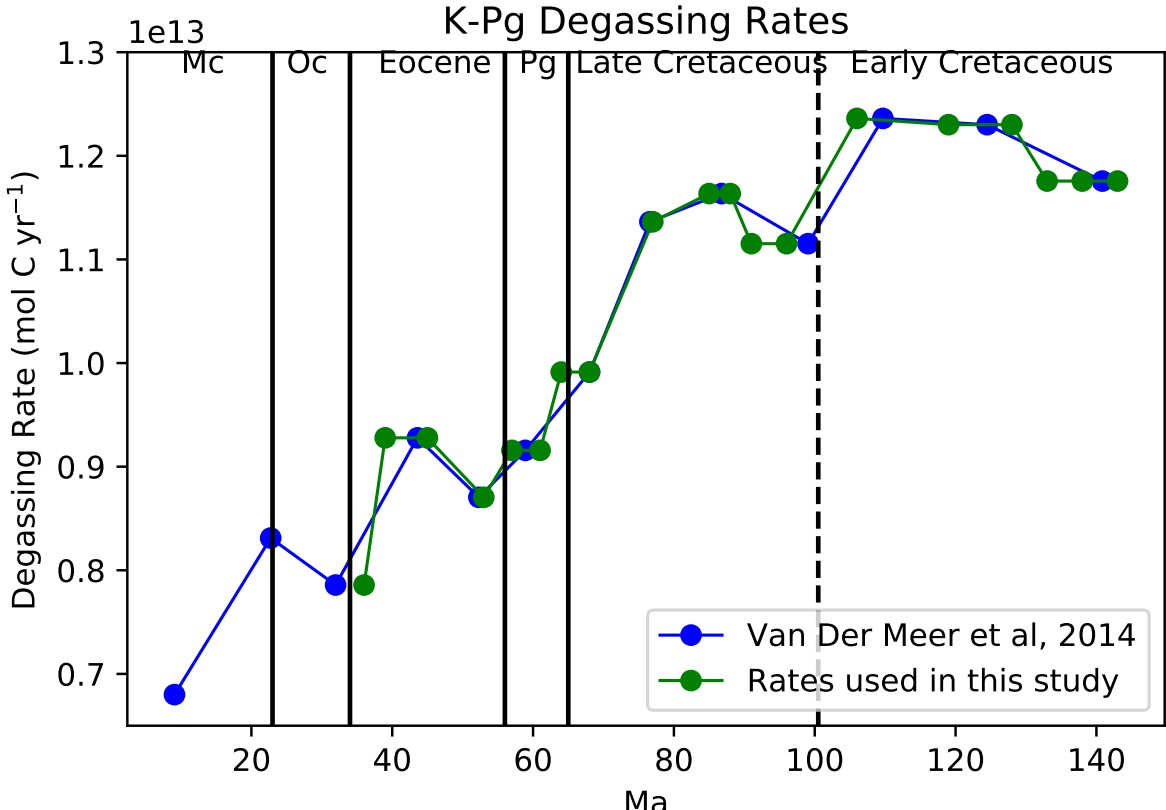

**Figure A1.** Degassing rates from Van Der Meer et al. (2014) (blue) used to provide variable degassing rates for the 19 time slices used in this study (green). Each time slice uses the rate from Van Der Meer et al. (2014) that is nearest in time. Degassing rates are highest in the early to mid Cretaceous (110-130 Ma) and gradually fall into the Paleogene with occasional rises.

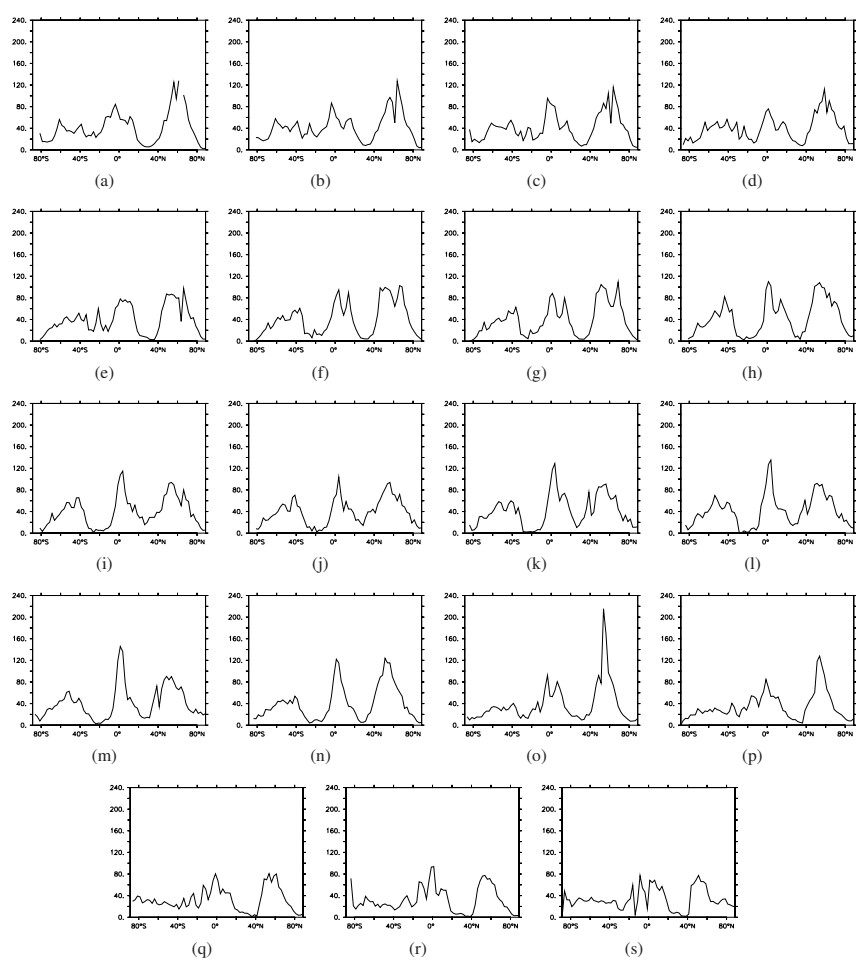

**Figure A2.** Modelled zonal mean runoff (cm yr$^{-1}$) from the earliest-Cretaceous (s) to the latest-Eocene (A). Zonal runoff means are low in the early-Cretaceous (s-p), but increase significantly into the mid-Cretaceous, especially in the northern mid-latitudes (o-k). Zonal mean runoff falls slightly into the late-Cretaceous, but there is an intensification in equatorial mean zonal runoff (j-h). In the Cenozoic, zonal runoff intensifies again (g-e) but begins to weaken again during the Eocene (d-a).