# Peer review of "Modelling the Impact of Palaeogeographical Changes on Weathering and CO2 during the Cretaceous-Eocene Period"

_EGUsphere, 2024_

## Author Comment (AC1)

**Review 1: Dana Royer**

The authors here investigate the role of paleogeography in controlling atmospheric $CO_2$ concentrations over the Cretaceous and early Cenozoic. The authors are at the leading edge of this kind of work, which previously coupled the GEOCLIM biogeochemical model with a low-resolution climate model FOAM. The major advance here is the use of a higher resolution climate model HadCM3L. Beyond the results specific to this study, the method is compelling and should be adopted by others.

I'll start with two comments:

- Why was van der Meer (2014) used for degassing instead of a newer, more sophisticated record like Muller (2022; https://doi.org/10.1038/s41586-022-04420-x) or Torsvik (2021; https://doi.org/10.1002/9781119528609.ch16)? Similarly, the Cenozoic $CO_2$ record of Foster (2017) has been usurped by Hönisch (2023; https://doi.org/10.1126/science.adi5177). This revised record is quite different, especially for the early Cenozoic. The problem, though, is how to suture this record to the pre-Cenozoic record of Foster (2017). I think the solution is to present both, with a disconnect at 66 Ma. I think it is important for readers to know that the pre-Cenozoic portion is less certain.

Concerning the use of the Van Der Meer et al. (2014) curve:
In recent years, we have been developing a radically new approach to modeling the carbon cycle in the geological past. This method uses climate models (GCMs) to simulate the weathering of continental silicate rocks. We have shown the importance of taking into account the spatial variability of temperature, and especially runoff, on the efficiency of weathering, and therefore on CO2 levels.

Without being an exhaustive intercomparison, our study is the first to explore the impact of the spatial resolution of climate simulations on calculated CO2 levels. As we focus on the role of paleogeography, we have selected the constant degassing rate simulations in the study by Goddéris et al. (2014), and compared them with our own simulations.

We added a simulation that includes variable degassing plus to show that the impact of CO2 degassing may be relatively limited. But this remains to be explored and is beyond the scope of our contribution. Unfortunately, it is not practical within the timeframe of the review process to re-run the models with Muller's degassing data. We can include some text in the manuscript to reflect the rationale behind exploring degassing mentioned here, and offer some speculation on the impact of using the Muller curve.

Regarding the CO2 history from proxies : I'm not sure what the word "usurped" means in this context. We think this must relate to a discussion between Hönisch and Royer. In any case, we are not involved in these discussions. We can use Hoenisch's $CO_2$ curve, which is published and therefore valid.

We suggest altering Figure 1 to include the Hoenisch curve in combination with the existing curve from Foster et al (2017). With regards to the degassing curves, we can include some text in the manuscript to reflect the rationale behind exploring degassing mentioned in this response and offer some speculation on the impact of using the Muller curve. Hopefully that should strike a balance between practicality and an acknowledgement of the Muller and Hoenisch data.

- Paragraph from lines 241-250: This analysis would be more compelling if the associated figure was a cross-plot of runoff vs. silicate weathering (but connect time-adjacent points with lines so that the time evolution can also be evaluated). Statements like "strengthens", "weakens", and "low confidence" need statistics to back them up.

I think the trends should be visually clear from the existing plots, but we can certainly include the cross-plots and statistics in the supplementary information.

Finer-level comments:

Line 12: "switch" shouldn't be in quotes; the phrase "global weatherability appears to switch regimes. In the Cenozoic," can be removed without affecting the intent of the sentence.

Agreed. We will make the suggested change.

Line 24: Weathering and burial of organic matter is important too.

In the present study, we focus on silicate weathering, as we believe that this highly complex process remains a major source of uncertainty in deep-time reconstructions of atmospheric $CO_2$, but we can certainly amend the text to acknowledge the role of carbon burial, especially as Reviewer 2 and the Editor have also made similar comments.

Section 1.1: Somewhere in here mention that two additional factors that are held constant are the organic subcycle (burial / weathering of organic carbon) and plant evolution (especially the evolution of angiosperms during the Cretaceous). Plant type is included in HadCM3L, but the direct effect of plant type on chemical weathering in GEOCLIM is not (based on my understanding).

The reviewer is right regarding the absence of a direct effect of plants on weathering in the version of GEOCLIM that has been used, so we can update the text accordingly.

My understanding is that there is still considerable disagreement in the community about the relationship between plants and weathering, so I would prefer not to linger too long on this topic given the lack of constraint in the relationship.

Line 62: Why do you say "Another" here? The previous paragraph (especially line 58) already introduces this topic.

We'll remove the word as it appears to be superfluous.

Figure 1: "Pg" (= Paleogene) is not correct; this should be Paleocene.

Thanks for spotting this, we'll update it accordingly.

Figure 3: It's odd (to me) to see the panels organized from young to old (instead of old to young). We have been culturally conditioned to read from left to right and to tell stories in chronological order. (The same is true for the time axis in Figures 1, 2, etc...).

For figure 3, I believe this is consistent with other palaeoclimate modelling studies, though I accept this may be a matter of personal preference for the respective authors.

For the other timeseries figures, my rationale for presenting them in this order is that values are usually presented in order of increasing value, which feels a little unnatural to me in a timeseries. It ultimately is a trivial change to implement, so I am happy to change this if the editors believe doing so would be more consistent.

Lines 234-236: I'm confused by this statement: shouldn't you be correlating runoff to $CO_2$ directly?

The intent of this statement was to say that initial silicate weathering fluxes are strongly correlated with runoff, and that when initial silicate weathering fluxes are high this results in lower steady state CO2 concentrations. I avoided including a direct correlation here as (in my mind), it is a multi-stage process (high runoff results in high silicate weathering, resulting in lower steady state CO2). I believe we also have correlations between runoff and CO2 directly, so those can be included as well.

**Review 2 : anonymous**

The authors investigate the influence of weatherability on long-term CO2 trends during the Cretaceous-Eocene period (145-34 Ma). To this end, they use climate model output (temperature and runoff) from 19 GCM simulations from HadCM3L as input for global climate and biogeochemistry model GEOCLIM. They find that palaeogeography strongly controls weatherability (primarily via changes in runoff) and that localised changes in weatherability can have an impact on global CO2 values, making a case for spatially resolved simulations of weathering-driven CO2 changes in Earth's history.

This is an interesting and solid paper which deserves publication after major review. My main concerns relate to a somewhat selective discussion in some places, a lack of depth in the analysis, and a lack of discussion of potential sources of uncertainty in the models.

Comments:
Lines 59-61: It should at least be mentioned that there is almost certainly also a contribution from CO2 drawdown by coal formation contributing to Carboniferous cooling.

We will mention the possible role of the organic carbon burial and oxidation of the old sedimentary organic carbon.

Line 85: The statement that "paleogeographies are relatively well constrained" might be true for the time period investigated in this paper but certainly not in general, so please clarify.

We will add "… for the last 150 million years.

Line 131-133: I understand that you focus on terrestrial processes here but saying that a "range of biogeochemical processes are modelled within the ocean layers" is really a bit vague for a scientific paper. Also, are there any feedbacks of these marine processes relevant for the results you are presenting in the manuscript?

The GEOCLIM model includes a 9 box-model for the ocean. The description of the oceanic model can be found in Goddéris and Joachimski (2004) and Donnadieu et al. (2006). The updated present version is fully detailed in a paper that we just submitted to Geoscientific Model Development, and it should be available online very soon (Maffre, P., Goddéris, Y., Le Hir, G., Nardin, É., Sarr, A.-C., and Donnadieu, Y.: GEOCLIM7, an Earth System Model for multi-million year evolution of the geochemical cycles and climate, Geosci. Model Dev. Discuss. [preprint], https://doi.org/10.5194/gmd-2024-220, in review, 2024.).

This said, there is no need to worry about the oceanic module of GEOCLIM in the present work, hence we only briefly mention the oceanic processes. Indeed, we ran two sets of simulations: one with prescribed atmospheric CO2 where we explore only the corresponding flux of CO2 consumption by continental silicate weathering. And in the second set the full GEOCLIM model is run until a steady state is reached. When the model reached the steady state, the global silicate weathering is equal to the prescribed solid Earth degassing. And the carbon cycle, including the fate of carbon within the ocean and sediments, adapts itself to keep the balance of the carbon and alkalinity at steady-state. As such, the oceanic module shouldn't be relevant for the results presented here. We can update the text accordingly to reflect this and make this clearer if needed.

Lines 171 & 181-183: Is there any impact of uncertainties in the vegetation model on the results, e.g. by using present-day PFTs?

This question is extremely difficult to answer. The PFTs defined for today are certainly not the same as those that existed 150 million years ago. But it's impossible to define the vegetation parameters in deep time. Certainly, the further back in time we go, the greater the

uncertainty. How can we determine the realism of the vegetation model given the absence of precise paleogeographic reconstructions of vegetation distribution based on field data (see, for example, the maps produced by Willis and McElwain, 2002). That said, when the number of PFTs is limited, vegetation models reproduce fairly well the spatial distributions of major biogeographical areas (desert, temperate, polar...), and this is probably the most important point (Donnadieu et al., 2009).

Line 172: Since runoff is crucial to your results and your arguments, it might be important to give some more details here - and to discuss the implications of uncertainties in modelled runoff on your results!

Runoff is at the same time the medium allowing rock weathering, and the carrier of the weathering products to the oceans. Consequently, it is not surprising that it plays a first-order role in the global weathering modulation.

It is well known that climate models are doing a better job modelling air temperature than continental runoff. Two factors are critical: the vegetation distribution and the continental relief. The first factor (vegetation) is calculated by the biospheric model couple to the GCM, while the second factor (relief) is not well constrained by geological and geochemical data. The present study requires many heavy numerical simulations. It is not practical to run numerous simulations, which would be beyond the scope of the study. The present study is by itself a sensitivity study where the 3D climate model FOAM (low res) has been replaced by a more spatially resolved model (HadCM3L). But the paleogeographic maps are not the same, the continental distribution differs the location of the main mountain ranges as well as their altitude differs. Even the total continental surface is not the same.

We can mention these limitations within the text. However, analysing inherent uncertainties in runoff within GCM models (and between different GCM simulations) would be a complex process in and of itself and we would argue that is well beyond the scope of this study, and probably better placed as a study of the GCMs themselves. We believe the key point of this study is to demonstrate the sensitivity of weatherability to runoff (and how changes in continental configuration can drive changes in runoff) rather than a "perfectly" accurate reconstruction of past $CO_2$ concentrations.

Lines 191-192: It might be unclear to the reader how you fix the $CO_2$ concentration in the model given a fixed degassing rate but a variable weathering rate? Are you referring to the climate model part relating to weathering?

This point is probably unclear as written. We run two sets of simulations:
Set 1: atmospheric $CO_2$ is fixed at 2.85x modern $CO_2$. Runs calculate the corresponding silicate weathering pattern. Of course, the global silicate weathering will change from one timeslice to the other. And so does the solid Earth degassing rate, the value of which will be equal to the solid Earth degassing. It is a convenient way to calculate the impact of paleogeography on the weathering flux. These runs do not require the integration of the full GEOCLIM model, but only of the weathering module.

Set 2: this second set of runs allows us to calculate the steady-state CO2. We agree that the sentence describing this set is awkward and it will be corrected. In those simulations, the solid Earth degassing can be fixed or made variable. The atmospheric CO2 will adapt until the global silicate weathering will compensate for the solid Earth degassing.

Lines 198 & 202: This might be confusing because you ran two sets of 19 simulations each. It becomes clear from the context but should be explained in a better way to avoid unnecessary confusion.

This will be clarified (see above)

Figure 1: Although the symbols are different, I think it is not a good idea to have both the CO2 proxy record and one of the sets of model simulations in red – there are many more colours out there…! ;)

There is some discussion in relation to comments from Reviewer 1 about whether to include the variable degassing rate (given that the impact seems relatively limited). Regardless, this is a reasonable comment and if the values are included, I will alter the symbols to be blue squares for consistency with the modelled data.

Caption of Figure 1, lines 2-5: In my opinion, figure captions should be for descriptions rather than discussion of results/comparison with literature. This should be moved to the text.

This is discussed later in the text but was included in the caption here for context. Happy to remove if editors agree.

Lines 209-210: This might be slightly confusing because you are discussing the set of simulations with fixed CO2 here. I think this can be explained better in the sense that you could say that you are comparing the two sets of simulation here…

Agreed – we will update to reflect this.

Caption of Figure 2, line 2: Again, this sentence might be better placed in the main text. Also, I am not sure I understand the basis for this assessment from the plotted lines.

Both temperature and runoff show some similarities to the weathering flux, but the relationship between runoff and temperature is more consistent (this is explored in the discussion). We can also include additional plots in the supplementary information showing the relationship between weathering and temperature if that would be helpful.

Figure 3: It might be useful to have some sort of a grid, at least in latitude, to be able to judge, e.g., where the ITCZ or the subtropical high-pressure belt would be expected etc.

I would tend to agree, however given the nature of the figure (a number of small subplots) I think a grid could obscure some of the maps. Increasing the size of the plots would necessitate a multi-page figure and my understanding is that such figures should be avoided. Additionally, there are plots of zonal mean runoff for each simulation in the supplementary information which should hopefully show how runoff changes with latitude (Figure A2). Given these restrictions I will avoid including a grid at this stage, but should it be requested by the editors we can re-plot the figures with a grid.

Discussion section: There is no discussion on the influence of model uncertainties on the results, see the examples mentioned above and below. This needs to be added in the revised version.

Please see our response to the aforementioned comments.

Lines 241-242: This sentence lacks precision and is therefore confusing. Which "mean"? Which "runoff rates"? Also, the total global runoff is shown in Figure 2, so why are you referring to Figure 4?

I'm not entirely sure what the reviewer is referring to here, but my understanding of the comment is that the sentence was unclear about whether the runoff rates were referring to those within GEOCLIM or from with HadCM3L. I will alter the text to clarify that these are referring to the values from GEOCLIM. The reference to Figure 4 will be amended to be for Figure 2.

Line 255, "climate patterns": This is a bit vague...

I will alter this to "climatic variables", which should be more specific within the context of the sentence.

Lines 258 and 264-265, "may be": I find this choice of words a bit strange given that you should have the model diagnostics to figure out what is going on in the models...

This was intended to suggest that while this is what the models indicate, it may not necessarily be the case in the "real world" but I understand how this may appear confusing here. We will remove "may be" to avoid potential confusion.

Lines 279-299 & 375: So you find agreement for (roughly) 145–125 Ma, 90–70 Ma, and 50–45 Ma or 45 Myr or less than half out of the total 111 Myr spanned by your simulations. I find it scientifically somewhat questionable to discuss the agreement during these time periods in detail without discussing the other time periods where the comparison fails. I think that the framing in the Conclusions (lines 411-413) is alright but this is not quite reflected in the Discussion.

This is a good point. We'll update the discussion to reflect the framing in the conclusion.

Figure 5: I think the vertical axes need better labels explaining that A shows a CO2 difference and B a difference in silicate weathering flux.

Agreed. The axis labels will be edited to reflect this.

Section 4.4: I am really not happy to call a fictitious CO2 trend based on model simulations with strong assumptions and disagreeing with proxies for a significant portion of the time period covered a "CO2 record".

Will change to "modelled [CO2] concentrations" where applicable.

Appendix A: I was rather surprised to discover in the appendix that climate variables were extrapolated towards lower CO2 levels and frankly shocked that this was not mentioned in the Methods section of the manuscript. What are the implications for your results? Have you at least run a test simulation making sure that the extrapolation method produces reasonable results?

I'm not sure I've fully understood the comment. I will try to elaborate but I apologise if this is not what the comment was referring to.

This was not mentioned in the main text as the "lower" values (i.e. below 280ppm) are associated with the original FOAM model, which we do not use for the simulations here. Additionally, all of our simulations (except for one run under the variable degassing rate) are well above modern CO2 values and fall within the 560-1120ppm range that the HadCM3L GCM simulations were run at. The only value that may be an issue is the final value at 34Ma under the variable degassing rate, but that value is still above 400ppm, and does not affect the overall trend seen in our simulations.

Indeed, Figure 5 examines the impact of using the extrapolation process on the original FOAM inputs (used in Godderis et al, 2014) (Extrap FOAM temp and Extrap FOAM runoff). The impacts on initial silicate weathering fluxes and steady state CO2 are an order of magnitude smaller relative to the impact of switching to using the HadCM3L climate inputs. The impact of transitioning from FOAM to HadCM3L is explored in detail in Hayes (2019) (https://research-information.bris.ac.uk/files/219744470/Final_Copy_2019_10_01_Hayes_N_PhD.pdf). If this unclear as written, we can expand on the relevant section to clarify the process.

Minor comments:
There are a few minor typesetting issues in the manuscript, e.g. missing spaces (e.g. lines 10, 16, 142), unclosed brackets (e.g. in the caption of Figure 3), messed up citation styles (e.g. lines 147, 153) which should be corrected.
Line 141: "Arrhenius"
Line 175: "palaeogeographries"
Line 235: Add "times"?

These will be corrected.

Line 271: Is "pure" the best word here...?

Will edit to "full GCM studies". This is hopefully a better phrasing to reflect that we are not running a GCM directly in this study, rather the outputs.

Somehow the heading "References" appears twice, including a blank page.

The double Reference heading has been fixed, although the blank page remains. This may be an issue with the LaTeX template that can be fixed during the typesetting phase.

**Editor Comments:**

The authors employ a coupled global climate and biogeochemical model over relatively high temporal and spatial resolution to assess the impact of paleogeography on weathering-controlled pCO2 during the late Cretaceous-Eocene.
This is an interesting study that merits publication after reviewer comments are addressed and appropriate revisions are made. The reviewers provide valid and constructive comments. I add here only a few additional comments to consider.

23-25— what about long-term burial of organic carbon, such as that of the late Paleozoic (for terrestrial Corg) or other times of substantial burial of marine Corg?

This has been raised by both reviewers – please refer to our response to Reviewer 2's first comment.

30-34— Others have suggested the importance of lithology on weatherability, e.g. exposure of mafic rock types (e.g. ophiolites) in orogenic belts.

We mention lithology on line 27, as well as later in the manuscript in relation to palaeolithologies. We can include a more explicit example of the role of lithology and weatherability in past if that would be helpful here (such as Donnadieu et al, 2004).

50-60— In this paragraph, it seems relevant to cite some of Cin-Ty Lee's (et al.'s) arguments re potential forcing from Cretaceous arc volcanism, for example. (E.g. Lee and Lackey, 2015; Lee et al., 2015 EPSL; Lee et al., 2016, Science).

We'll include the suggest references.

70— Weathering is also affected by erosion rates; see, e.g. Bufe et al., 2024, Science.

This is acknowledged on lines 73 and 152. We don't dwell too much on erosion as we don't include it in our model for consistency with Godderis et al (2014), but I'm happy to include the reference to Bufe et al (2024) and briefly expand on weathering and erosion.

96-100— Re the effect of supercontinents vs dispersed continents on pCO2— in detail, it seems a bit more complex. Eg, consider that late Penn-early Permian Pangaea was

assembled into Pangaea, but CO2 values were quite low (for reasons that remain somewhat debated in detail), whereas mid-late Permian Pangaea was both arid and characterized by high pCO2 (for reasons of both volcanic outgassing and equatorial aridity). In other words, both times are characterized by supercontinent configuration— early vs late in the process— but exhibit contrasting pCO2 values.

This is a valid point and it is indeed more complex. However, I believe it is accurate during the time period we study here (additionally, by the cretaceous the breakup of Pangaea is well underway). We mostly attribute the changes in weatherability to changes in runoff associated with continental breakup, but we do mention elsewhere in the paper that changes can occur within a supercontinental configuration due to runoff (lines 103, 329, 331). Again, we can refer to this more explicitly within the text if that is helpful.

102—should be "draw *down*"

This will be corrected.

175— should be "palaeogeographies"

This will be corrected.

253— Also depending on orography; e.g. equatorial Pangaea's position in the rain shadow of the Central Pangaean Mountains (albeit paleoaltimetry is extremely difficult to constrain).

We mention the role of orography elsewhere in the text but we will include a more direct reference in the context mentioned here.